# Induction of Accelerated Aging in a Mouse Model

**DOI:** 10.3390/cells11091418

**Published:** 2022-04-22

**Authors:** Nanshuo Cai, Yifan Wu, Yan Huang

**Affiliations:** MOE Key Laboratory of Gene Function and Regulation, Guangzhou Key Laboratory of Healthy Aging Research and State Key Laboratory of Biocontrol, School of Life Sciences, Sun Yat-sen University, Guangzhou 510275, China; cainsh@mail2.sysu.edu.cn (N.C.); wuyf97@mail2.sysu.edu.cn (Y.W.)

**Keywords:** aging, aging-related disease, mouse model

## Abstract

With the global increase of the elderly population, the improvement of the treatment for various aging-related diseases and the extension of a healthy lifespan have become some of the most important current medical issues. In order to understand the developmental mechanisms of aging and aging-related disorders, animal models are essential to conduct relevant studies. Among them, mice have become one of the most prevalently used model animals for aging-related studies due to their high similarity to humans in terms of genetic background and physiological structure, as well as their short lifespan and ease of reproduction. This review will discuss some of the common and emerging mouse models of accelerated aging and related chronic diseases in recent years, with the aim of serving as a reference for future application in fundamental and translational research.

## 1. Introduction

The world’s population continues to grow, with those over 65 being the fastest growing age group. In 2019, one out of every eleven people in the worldwide population was over the age of 65. By 2050, this proportion will increase to one in every six people over the age of 65. In some regions, such as in Europe and North America, one in four people will be over 65 years old. Aging is a key risk factor for multiple disorders. More than 75% of the elderly suffer from at least one chronic disease, and the problem of “unhealthy longevity” for the elderly is prominent [1,2].

As countries face the challenges of aging populations, there is a need to promote healthy aging and provide adequate social protection. To promote a healthy aging process and prevent aging-related health problems, correctly understanding aging mechanisms and developing effective and affordable intervention strategies for anti-aging have great social significance and huge economic benefits. During this process, an animal model of aging is a powerful tool for us to study the mechanism of aging.

Generally, aging models are divided into two categories: natural aging models and accelerated aging models. In the process of aging, naturally aging mice develop many phenotypes similar to normal human aging like cataracts [3] and muscle weakness [4]. For example, the most well-known strain of mice, C57BL/6, have a lifespan of two to three years, while naked mole-rats can live up to 30 years. One longevity mechanism of naked mole-rats is their effective DNA damage repair system. However, research on natural aging model is time-consuming, labor-intensive, and expensive, with large individual variations compared to the accelerated aging models [5,6]. As such, the induced accelerated aging model is favored by researchers because of its convenient source, short modeling time, and relatively controllable aging effect. For aging and aging-related chronic disorders, mice have become one of the most important animal models for studying various aging-related human diseases due to their similarities to humans in terms of genetic background and the structure and function of various organs or systems, as well as the advantages of short lifespan and ease of reproduction [6,7]. With the help of mouse models, researchers have revealed the mechanisms of aging as well as the pathogenesis of various chronic diseases, and many therapeutic approaches for such chronic diseases have been validated in mouse models preclinically.

In this review, we discuss some common and emerging mouse models of accelerated aging and its related chronic diseases in recent years, with the aim of serving as a reference for future applications.

## 2. Systemic-Induced Accelerated Aging Mouse Model

In this section, we will review some commonly used mouse models of accelerated systemic aging including drug treatment, genetic engineered models, irradiation induction, etc. They are characterized by an aging phenotype in multiple tissues or organs, reflecting systemic aging. The systemic-induced accelerated aging mouse models are summarized in Table 1.

### 2.1. The D-Galactose-Induced Senescence Model

D-galactose is a common aldohexose that exists naturally in the body and in daily foods [8]. After ingestion, a healthy adult can metabolize and eliminate a maximum daily dose of 50 g of galactose from the body within about eight hours [9]. However, when galactose accumulates to high levels, reactive oxygen species (ROS) are generated by mitochondrial respiratory chain enzymes, xanthine oxidase, lipoxygenase, cyclooxygenase, nitric oxide synthase, and peroxidase. Increased ROS can subsequently lead to elevated oxidative stress and inflammation, inducing mitochondrial dysfunction and apoptosis [10]. Meanwhile, the elevated mitochondrial ROS level can lead to the activation of many biochemical pathways, such as the polyol pathway, the formation of advanced glycation end products (AGEs), the activation of protein kinase C, and the hexosamine pathway [11,12]. Overall, D-galactose-induced methods can increase aging markers such as AGEs, receptors for advanced glycation end products (RAGEs), aldose reductase (AR), sorbitol dehydrogenase (SDH), decreased telomerase activity, shortened telomere, β-site amyloid precursor protein cleaving enzyme 1 (BACE-1), amyloid β (Aβ), aging-related pathways (p16, p21, p53, etc.), and positive senescence-associated β-galactosidase (SA-β-gal) staining [13]. Multiple tissues and organs, including the brain, heart, lung, liver, kidney, reproductive system, gastrointestinal system, and so on, manifest aging phenotypes after D-galactose treatment [14].

Previous works have shown that D-galactose induces brain aging by increasing mitochondrial dysfunction, oxidative stress, inflammation, apoptosis, and decreasing the expression of brain-derived neurotrophic factor. D-galactose injections may induce brain aging similar to human brain aging in many ways, including mitochondrial dysfunction, increased oxidative stress, decreased ATP production, neuronal degeneration and apoptosis, and cognitive deficits [13,15,16,17]. D-galactose increases the neuro-inflammation markers via activating NF-κB, leading to memory impairment [18,19]. Besides learning and memory inhibition, D-galactose-treated mice also exhibit depressive and anxious behaviors [20].

The leading cause of death in elderly people worldwide is cardiovascular disease [21]. D-galactose treatment increases the risk of cardiovascular disease, which is associated with excess ROS and oxidative stress. Persistent oxidative stress has been revealed to be related to decreased ferric reducing antioxidant power and lower activity of Cu-Zn superoxide dismutase, leading to myocardial damage [22]. Studies have shown that galactose reduces endogenous hydrogen sulphide producing enzyme cystathionine γ-lyase (CSE) [23] and antioxidant enzymes such as catalase (CAT), haem oxygenase-1 (HO-1), superoxide dismutase (SOD), glutathione peroxidase (GSH-Px), and nitric oxide synthase (NOS), leading to decreased total antioxidant capacity and inducing lipid peroxidation markers including malondialdehyde (MDA), lipid hydroperoxides (L-OOH), and conjugated dienes (CD) in cardiac tissue [24,25]. D-galactose increases whole heart weight and left ventricle weight, which is associated with hypertension and aging. At the same time, the heart tissue showed enlarged myocardial fibers, blurred structure, shortened distortion, widening of the interval, and obvious capillaries of myocardial interstitial congestion [26]. D-galactose treatment resulted in cardiac fibrosis, significant accumulated collagen, and disordered arrangement of fibroblasts compared with the control. D-galactose also increased cardiac apoptosis markers [27]. Excessive D-galactose can be transformed to advanced AGEs via the Maillard reaction [28]. AGEs bind to the receptors, RAGE, increasing ROS production via NADPH oxidase. NADPH oxidase further activates p38 MAP kinases, causing transcription factor NF-κB translocating into the nucleus, where transcription of inflammatory cascades like tumor necrosis factor alpha (TNF-α) are enhanced [29]. D-galactose treatment can also increase fibrotic markers such as connective tissue growth factor (CTGF), transforming growth factor β1 (TGF-β1), phosphorylated mitogen-activated protein kinase 1/2 (p-MEK1/2), phosphorylated extracellular signal-regulated kinase 1/2 (p-ERK1/2), matrix metalloproteinase (MMP), and pathological specific protein 1 (SP1) [30].

D-galactose treatment also increases oxidative markers (e.g., MDA) and decreases antioxidant enzymes (e.g., SOD, GSH-Px and NOS) in lung, liver, and renal tissues [26,31,32,33,34,35]. D-galactose treatment can affect lung elastic constitution. The primary effects of treatment on the lungs are increased alveolar size and reduced elastic recoil, which may facilitate airway closure [36]. D-galactose treatment is considered to successfully mimic the natural aging process by increasing oxidative stress, fibrotic status, and chronic inflammation in the lungs.

The liver is the main site where D-galactose is metabolized, thus excess D-galactose in the body may significantly affect the liver. As previously mentioned, high levels of D-galactose can react with free amines in the amino acids to form AGEs, which is found to be involved in the progression of various liver diseases [12]. High levels of D-galactose lead to the accumulation of its final metabolite, galactitol, which will eventually lead to ROS accumulation through the p38 MAPK/NRF2/HO-1 signaling pathway and cause cellular osmotic stress in the liver [37]. D-galactose treatment significantly increased apoptotic proteins including Bax, procaspase-3, and caspase-3, and raised the ratio of Bax/Bcl-2 in the liver tissue [34,38,39]. Various changes similar to natural aging were also observed in the D-galactose-treated livers. Compared with controls, the livers of D-galactose-treated animals had lower levels of glycogen and more lipid deposition. Masson’s trichrome staining showed obvious collagen fibers around blood vessels and in the interphase of liver tissue, and irregular pseudo-lobules around liver tissue, indicating a state of liver fibrosis [40,41].

D-galactose treatment significantly reduced the renal index of animals, and markers of acute kidney injury such as uric acid and cystatin C (Cys-C) were also increased [42]. In the kidney, D-galactose administration resulted in an increase in the TUNEL-positive cells and DNA fragmentation. In addition, p21 expression and the staining intensity of SA-β-gal were also increased in kidney cells [43]. Extensive glomerular and tubular damages were detected in D-galactose-treated animals, as the number of tubules with cellular necrosis from the renal cortices and outer medulla were significantly increased [43,44].

In the male reproductive system, D-galactose induced oxidative stress, marked by an increase in MDA levels in the prostate, testis, epididymis, and decrease in SOD activity in the testis. Peroxidation in the testicular and epididymal mitochondrial fractions was also significantly increased after D-galactose treatment [45]. Female reproductive aging is characterized by decreased levels of estrogen, progesterone, inhibin B, anti-Müllerian hormone, and androgens, which include free testosterone, dehydroepiandrosterone (DHEAS), and androstenedione [46]. Compared to the control groups, D-galactose administration produced aging-associated changes like reduced estrogen and progesterone levels and increased FSH and luteinizing hormone (LH) levels. Ovarian follicle regression and atrophy on the uterine wall and endometrial gland were observed after D-galactose treatment, which indicates disrupted estrous cycles and damaged uterine and ovarian tissues [47].

D-galactose injection can lead to changes in the level of oxidative stress that affect the microbial environment in the intestine and lead to intestinal flora disorder [48]. The ecology of intestinal flora is closely related to aging, and intestinal ecological disturbance can lead to accelerated aging and a shortened lifespan [49]. Transferring gut microbiota from aged to young germ-free mice triggered innate immune and inflammatory responses. Effects of aging include increased differentiation of CD4^+^ T cells in the spleen, upregulation of intestinal inflammatory cytokine such as TNF-α, and increased cycling of bacterial-derived inflammatory cytokines [50,51]. In addition, D-galactose administration significantly decreased the small intestine propulsion rates and prolonged gastrointestinal transit time [49]. The aging gut triggers chronic inflammation, leading to gut dysplasia and intestinal dysplasia in turn leads to defective epithelial function, predisposing the host to infection, neoplasia, and increased mortality [52].

### 2.2. Senescence-Accelerated Mouse/Prone

Researchers from Kyoto University found an aging phenotype from a subset of pups while maintaining an inbred line of AKR/J mice. Characteristics of these mice include hair loss, reduced activity, and a shortened lifespan. These aging traits are thought to develop as a result of elevated oxidative stress and are inherited by their offspring. Further, the accelerated aging mice were grouped into several distinct subtypes according to their phenotypes [53].

Senescence-accelerated mouse/prone (SAMP) is a group of inbred mouse strains that typically exhibit accelerated aging [54]. Meanwhile, since it shows various aging-related diseases similar to humans, such as aging amyloidosis, senile osteoporosis, learning/memory impairment, etc., in specific lines, it is widely used for aging research. Cellular senescence in various cell types, including astrocytes, endothelial cells, progenitor cells, retinal epithelial cells, and fibroblasts, was found in the aging SAMP mice [55,56,57]. SAMP mice exhibit an increase in ROS generated by mitochondria or other cellular sites, which not only causes damage to mitochondria, but also triggers degradation that leads to the aging outcome [58].

SAMP1 mice are characterized by aging amyloidosis, immune dysfunction, renal atrophy, hearing loss, and senile pulmonary hyperinflation [59,60]. The lifespan of the SAMP1 mice is about 40% shorter than senescence-accelerated resistant mice (SAMR1), and various signs of aging appear early in appearance.

The SAMP6 mice model is a senile osteoporosis model animal that tends to develop osteoporosis at an early stage with aging due to its low bone density in instar [61]. Bone marrow transplantation experiments have revealed that the cause of SAMP6 osteoporosis is abnormalities in bone marrow stem cells [62]. The incidence of spontaneous leg fractures due to osteoporosis is high in adult SAMP6 mice. Cellular senescence in myeloid progenitors disrupts their differentiation in favor of adipogenesis over osteogenesis. This mechanism is thought to contribute to low osteoblast activity and osteoporosis in SAMP6 mice and the elderly [61,63,64].

The SAMP8 mice develop age-associated deficits in learning and memory and also exhibit various age-related neuropathological changes similar to aging humans [65,66]. Neuropathological changes including astrogliosis, microgliosis, and neurodegeneration occur as early as five months of age [67]. SAMP8 mice also showed accumulated amyloid and age-related microtubule-associated protein tau (MAPT) hyperphosphorylation, as well as increased nitric oxide synthase activity, further demonstrating their feasibility as a brain aging model [68,69]. In addition, decreased activities in SOD, CAT, glutathione reductase, and GSH-Px, and increased activity in acyl-CoA oxidase were detected in SAMP8 mice at 1–12 months of age [70]. SAMP6 and SAMP8 mice also develop many other age-related diseases, including retinal degeneration, testosterone deficiency [71], myocardial fibrosis [72], and hepatic lipid deposition [73].

The SAMP10 mice exhibit learning and memory impairment with aging, and atrophy is observed in the cerebral cortex and limbic system, so it is considered to be a spontaneous model animal for aging and brain degeneration. The atrophy of the frontal cerebral cortex and olfactory bulb is marked in SAMP10 mice [74,75].

So far, more than ten strains of SAMP mice have been identified and widely used in aging studies, each of which can develop various age-related diseases such as renal fibrosis (shrinking of the kidneys), immune dysfunction, and degenerative joint disease like osteoarthritis (OA), etc.

### 2.3. Rps9 D95N Mouse

Rps9 D95N is a ribosome ambiguity mutation that causes error-prone protein synthesis in mammalian ribosomes, resulting in increased error-prone translation. Rps9 D95N mutant mice exhibit features of accelerated aging, including morphological (altered fur, cataracts, and hunched posture), physiological (body composition and function, body weight, fat mass, and muscle strength), and pathological (shortened lifespan, mouse urinary syndrome and extramedullary hematopoiesis), which also complements and explains the link between accumulation of erroneous proteins resulting from protein mistranslation and individual aging [76].

### 2.4. Progeria Syndrome Mouse

Mouse models of progeria syndrome have emerged as an attractive tool for evaluating intervention strategies for unhealthy aging due to their short lifespan, relatively simple strategies by single gene deletion or mutation, and their strong phenotypic similarities to normal aging [7]. To understand important mechanisms of the aging process, progeria mice *(Lmna^−/−^*, *Wrn^∆hel/∆hel^*, *Csa^−/−^* or *Csb^−/−^*) from common progeria types such as Hutchinson-Gilford Progeria, Werner Syndrome, and Cockayne Syndrome are being widely used.

The most common phenotypes in mouse models of progeria can be observed in bones, joints, skin, nervous system, adipose tissue, skeletal muscle, cardiovascular system, liver, kidney, and the hematopoietic system. Less common lesions occur in the gonads, eyes, and occasionally in the gastrointestinal tract [77].

Some human progeria syndromes like Werner Syndrome exhibit osteoporosis. Current data from a mouse model of progeria indicate that senile osteoporosis is the result of reduced bone turnover and loss of bone mass due to defects in osteoblast progenitor cells, osteoblast differentiation, or osteoblast function [77]. Degenerative joint disease is another major symptom that affects the elderly. Mutation in the *Xpd* gene of nucleotide excision repair (NER) leads to a short lifespan, causing trichothiodystrophy (TTD) in humans. As expected, *Xpd^TTD/TTD^* mice exhibited a significant decrease in subchondral bone plate thickness compared to that observed in wild-type mice. Surprisingly, female *Xpd^TTD/TTD^* mice exhibited less cartilage damage and fewer lost articular cartilage, compared to WT females [78].

During the aging process, skeletal muscle was inevitably accompanied with a reduction in muscle mass, known as sarcopenia or age-related muscle atrophy, and macroscopic examination of sarcopenic mice will show weight loss and marked reduction in muscle mass. Sarcopenia is characterized by the loss of muscle fibers and smaller fiber cross-sectional area that is defined as fiber atrophy [79]. The progeria mouse model *Bub1b^H/H^* and *Bub1b^+/GTTA^* mice showed decreased mean muscle fiber diameter, increased myofiber size variation, increased intermuscular fibrosis, and impaired regenerative capacity in skeletal muscles [80].

Common brain aging diseases include Alzheimer’s disease (AD) and cognitive dysfunction syndrome. Brain atrophy, neuronal loss, neurofibrillary deposition of Aβ or senile plaques, intraneuronal tauopathies (neurofibrillary tangles, NFTs), cerebrovascular amyloid angiopathy, neuronal lipofuscinosis, vascular and meningeal calcification, decreased white matter integrity, and astrogliosis are common age-related neurological pathologies [81]. The brains of several mouse models of progeria show neurodegenerative changes at an early stage. Similar to age-related gliosis, *Bub1b^H/H^* mice have increased numbers of astrocytes and microglia at one and five months of age, respectively. Additionally, *Xpg^−/−^* mice developed more astrocytes and increased activation of microglia in the brain and spinal cord [82].

### 2.5. Mitochondrial DNA Polymerase Mutant Mouse

PolgA is the catalytic subunit of mtDNA polymerase encoded in the nucleus. Mice that were knocked-in proofreading-deficient *PolgA* developed a mutator phenotype with an over threefold increase in the mtDNA levels with point mutations and deletions. The mtDNA polymerase mutant mouse manifest reduced lifespan and premature phenotypes including weight loss, hair loss, reduced subcutaneous fat, kyphosis, osteoporosis, anemia, infertility, depletion of spermatogonia, heart hypertrophy, etc.

Besides, the mtDNA polymerase deficient mouse showed age-related loss of skeletal muscle likely contributing to sarcopenia, and age-related loss of spiral ganglion neurons, which is recognized as a feature of presbycusis.

Mutations in mtDNA may contribute to premature aging of the organism through apoptotic loss of critical, irreplaceable cells and thus induce tissue dysfunction. The *PolgA* mutation mice showed increased TUNEL-positive cells in intestinal epithelial, thymic, and testicular tissues. Compared to wild type mice, cleaved caspase-3 levels were significantly higher in the cytoplasmic fractions of viscera from *PolgA* mutation mice. Thus, mtDNA mutation-induced senescence was associated with the activation of apoptotic pathways in these tissues [83,84].

### 2.6. Total Body Irradiation (TBI) Model

Radiotherapy and chemotherapy are commonly used in anti-tumor therapy. However, they result in long-term damage to a variety of organ systems, including the cardiovascular system, gastrointestinal tract, lung, liver, musculoskeletal system, and neurological network. These treatments cause tissue phenotypes resembling accelerated aging [85,86]. Studies showed that sub-lethal total body irradiation (TBI) in mice induces progressive premature frailty and damage various tissues and organs. Radiation-induced frailty can be used to predict increased mortality and is associated with cognitive decline. Compared to control, TBI mice gradually developed frailty at a time approximately two times earlier and the frailty phenotype of irradiated mice was similar to those without irradiation at a higher age [87].

TBI may induce cellular and tissue damage by direct ionization effects and indirect free radicals generated during water radiolysis [88]. After TBI, changes of whole blood antioxidant capacity and red blood cell (RBC) glutathione (GSH) levels happened within two months in mice. The GSH levels and GSH/oxidized glutathione (GSSG) ratio in RBC decreased chronically after TBI ≥ 1 Gy [89]. Following a single and relatively low dose of TBI, the number and ability of hematopoietic stem cells (HSCs) to generate T cells is markedly and permanently impaired, thereby accelerating aging-related thymic involution [90].

Acute stressors such as radiation can cause senescent cells to accumulate in the elderly and animals. These senescent cells continue to secrete proteins that induce chronic inflammation, break tissue homeostasis, and disrupt organ function, a phenomenon called senescence-associated secretory phenotype (SASP) [91]. Individuals that received acute total body irradiation of approximately 1.5 Gy or more will suffer from bone marrow dysfunction and pancytopenia, the magnitude of which increases gradually with increasing intensity of irradiation [92].

After TBI of adult rats exposed prior to skeletal maturation, late degenerative joint damages in articular cartilage and trabecular bone were detected with a non-invasive imaging techniques [93]. In vitro, radiation exposure has been shown to cause acute degenerative alterations in the cartilage matrix composition and metabolism via lower proteoglycan (PG) content and compressive stiffness [94,95]. Radiation can also cause damage to the ovarian environment as well [96]. In adult female mice, drastic primordial follicle loss was observed in serial sections of ovaries even at the lowest dose of irradiation [97].

### 2.7. Ozone-Induced Senescence Model

The aging model produced by ozone damage can cause senescence in many tissues such as the heart, kidney, lung, and skin. Recent findings suggest that the ozone may drive neurological disorders such as cognitive decline, memory impairment, and AD symptoms [98]. Acute ozone exposure has effects on the nervous system of mice, causing reactive microgliosis and increased expression of Aβ in cortical and limbic regions of the brain [99]. Ozone exposure stimulates lung tumor growth and exacerbates the production of lung inflammation [100,101]. Lung epithelial injury caused by ozone exposure can lead to inflammation response, which results in the generation of multiple cytokines and chemokines and leads to neutrophil influx [102]. Studies indicate that TNF-α is required for ozone-induced airway hyperresponsiveness and inflammation, and the requirement appears to depend on the strength and duration of the ozone exposure [103]. Moreover, the antioxidants metallothionein (MT) and HO-1 are strongly induced by acute ozone exposure [104].

### 2.8. Chronic Jet-Lag Mouse

Jet-lag has repeatedly been shown to hasten death in animals [105].

When the core component of the circadian clock genes *Bmal1* was knocked out, *Bmal1**^−/−^* mice had reduced lifespan and displayed a number of symptoms of premature aging including sarcopenia, cataracts, less subcutaneous fat, organ shrinkage, and others. The early aging phenotype is associated with observed high levels of reactive oxygen species in certain tissues of the *Bmal1**^−/−^* mice [106].

A variety of research studies used rodents treated with chronic jet-lag by alternating the timing of light/dark cycles. The circadian rhythm regulates crucial cellular activities including metabolism and hormone secretion through circadian clock genes, which can modulate orchestrated physiological rhythms in tissues and the whole body [107]. Studies showed that the jet-lag-related dysregulation of the innate immune system is associated with altered or abolished rhythms in the expression of clock genes in the suprachiasmatic nucleus (SCN), liver, thymus, and peritoneal macrophages in mice [108]. Body temperature rhythm, corticosterone level in serum, and expression of mPER1 protein were significantly altered in the SCN in jet-lag mice compared to controls.

Dysregulation of the central clock genes led to increased c-Myc expression and enhanced cell proliferation and metabolic dysfunction and promoted growth and progression of tumors [109,110,111,112]. Chronic jet-lag increased the level of β-galactosidase staining in the liver as well as induced spontaneous hepatocellular carcinoma (HCC) and early breast cancer in mice [113,114,115]. Besides, chronic jet-lag during pregnancy can result in abnormal cardiac structure and impaired cardiac function in offspring [116].

Jet-lag can augment the risk for metabolic syndromes like obesity [117]. Jet-lag treatment induced more fat accumulation and significantly larger adipocytes, and also increased body weight and altered the metabolic gene profile in the mouse liver [118]. Furthermore, decreased microbial abundance, richness, and diversity in both feces and jejunal contents were observed in mice subjected to chronic jet-lag [119].

Jet-lag treatment aggravated depression-like behaviors as well as corticosterone-induced depression-like behavior. Jet-lag treatment decreased the mRNA expressions of telomere repeat binding factor 2 (*Trf2*) and telomerase reverse transcriptase (*Tert*) in the liver, hippocampus, spleen, and muscle. Moreover, jet-lag treatment significantly decreased the number of mitochondria, the level of NAD^+^, and the mRNA expressions of *COX1* and *ND1* [114]. Results from preclinical studies have suggested that the circadian system may play a critical role in drug addiction [120]. Interestingly, the expression levels of the *Cat*, *Gpx1*, and *Sod1* genes decreased more in the livers of mice subjected to both D-galactose and jet-lag, suggesting a synergistic effect of jet-lag and D-galactose on the aging process [121].

## 3. Tissue-, Organ-, or System-Specific Mouse Models of Aging-Related Diseases

Among the leading effects of ageing is the heightened incidence of various aging-related diseases, and mouse models continue to serve an essential part in the study of the pathogenesis and treatment of these illnesses. A variety of commonly used and emerging mouse models have been developed for different aging-related diseases, with the aim of reproducing as closely as possible the progression of the diseases in humans (Figure 1).

### 3.1. Model of Aging Brain or Nerve System

Although there are some genetic similarities between mice and humans, the two still diverge in the expression levels of certain age-related genes, which leads to potentially different aging process in the central nervous system of humans and mice [122]. Therefore, changing the expression levels of those specific genes that contribute to the aging process of the brain is the main strategy for modeling the aging brain in mice. For now, to ensure that experimental results can be extrapolated to humans, research on such models has mainly concentrated on evolutionarily conserved mechanisms that modulate aging. These conserved mechanisms include genomic instability, epigenetic changes, telomere attrition, mitochondrial dysfunction, loss of proteostasis, and dysregulated nutrient sensing [122,123].

For instance, as a neurodegenerative disease, AD represents one of the most common neurological disorders. The number of people affected by the disease has been recorded as over tens of millions worldwide, and the number will continue to rise. Also, it is the most common cause of dementia [124]. Over the past decades of research, with the help of animal models, substantial advances have been made, expanding our understanding of this disease.

So far, the etiology of AD is not yet completely understood. It is thought that genetic elements play key roles in the onset of such disease [125]. The typical histopathological features of AD are Aβ deposits and NFTs in the brain. However, wild-type mice do not spontaneously exhibit symptoms of AD [126]. Because AD-related proteins differ in sequence, pathogenicity, and number of isoforms between rodents and humans [127], the main strategy for modeling AD in mice is to construct transgenic mice that cause amyloid deposition or NFTs in the brain.

Common AD-associated mutant proteins in humans include Aβ, presenilin 1 (PS1), apolipoprotein E (ApoE), and MAPT. In the 1990s, the first transgenic AD model mice stably expressing the mutant human Aβ precursor protein (APP) were constructed [128]. After this, transgenic mice carrying multiple human AD-related mutant proteins emerged. One of the most commonly used AD models today is hAPP/PS1 lines, which carry both mutated human APP and PSEN1, including transgenic strains PSAPP, APPswe/PS1ΔE9, 5XFAD, and 2xKI [129]. Compared to monogenic lines containing only mutated APP or PS1, these transgenic mice exhibit earlier and faster onset of amyloid accumulation and cognitive impairment [130]. However, such AD models do not exhibit signs of NFTs, which can be imitated by mouse models that express human MAPT. Based on this, Oddo et al. constructed a 3xTg model combining the human APP, PS1, and MAPT mutations [131]. Since this model can show both Aβ deposition and NFTs in the brain, it is considered to be the well-established transgenic model of AD. Consistent with patients with AD, some of these transgenic mouse models (e.g., PS19, APPswe/PS1ΔE9) showed increased levels of NF-κB pathway-related proteins (IKKβ, p65, and COX-2) in the brain, indicating upregulation of brain inflammation in these mice. Therefore, the use of such mouse models will also help us to further clarify the relationship between neuroinflammation and the pathogenesis of AD [132,133].

### 3.2. Model of Aging Muscle

The aging of the body is accompanied by the aging of the skeletal muscles. Among other things, sarcopenia, which is a widespread progressive skeletal muscle disorder, is associated with an increased probability of adverse consequences like falls, fractures, physical disability, and death, and its risk increases with age [134]. Modeling of sarcopenia in mice is divided into two main categories: models of genetic engineering and chemical or dietary-induced models.

For genetic engineering models, the majority of research has employed knockout (KO) mice. For example, mitofusion2 (Mfn2) is one of the important protein components mediating mitochondrial fusion, and Mfn2 KO mice exhibit mitochondrial dysfunction in skeletal muscle cells and specific atrophy of type IIb glycolytic fibers [135]. Collagen VI, an extracellular matrix (ECM) protein, has a critical role in skeletal muscle. Six-month-old *Col6α1^−/−^* mice exhibit alterations of the diaphragm consistent with aged wild-type mice, such as abnormal tricarboxylic acid (TCA) cycle and decreased autophagy in diaphragm cells, indicating the *Col6α1^−/−^* mouse could be considered as a premature model of skeletal muscle aging [136]. Additionally, interleukin 10 (IL-10) [137], SOD1 [138], and NOD-like receptor protein 3 (NLRP3) [139] deficient mice have also been employed in studying the pathogenesis of sarcopenia as well as the intervention of therapeutically targeting such a disease. Overexpression of certain proteins can also lead to sarcopenia, such as TNF-α transgenic mice that exhibit reduced muscle mass, muscle fiber diameter, and Pax7^+^ muscle stem-cell content [140].

For chemical or diet-induced sarcopenia models, dexamethasone is a common inducer, which is essentially a glucocorticoid that triggers muscle atrophy in mice. It is shown that dexamethasone induces upregulation of ubiquitin ligases in muscle, including muscle atrophy F-box (MAFbx) and muscle ring finger 1 (MuRF1), which may further mediate the degradation of muscle atrophy-associated proteins [141,142]. In addition, diet-induced sarcopenia mouse models allow us to investigate the co-occurrence of sarcopenia with other disorders, e.g., sarcopenia can also develop from the reduction in muscle mass and strength caused by certain chronic diseases. Fabián et al. [143] treated mice with hepatotoxin 5-diethoxycarbonyl-1,4-dihydrocollidine (DDC), thereby inducing sarcopenia secondary to chronic liver disease (CLD), as evidenced by reduced muscle strength and motility, as well as the reduction in muscle fiber size and its type of transformation in mice.

### 3.3. Model of Aging Heart

Heart failure is a complex disease that can eventually result from almost all cardiovascular disorders, like myocardial infarction, atherosclerosis, and hypertension. Based on the left ventricular ejection fraction, heart failure is clinically classified into two major categories: heart failure with reduced ejection fraction or preserved ejection fraction (HFpEF) [144]. Among them, due to the increasing morbidity and mortality of HFpEF in recent years and the lack of effective therapeutic options for this disease, research on HFpEF has received increasing attention and as a result, some mouse models of HFpEF have been developed.

Long-term infusion of angiotensin II (ANGII) into mice based on a mini-osmotic pump is one of the most common methods of modeling HFpEF. Elevated levels of ANGII in the mouse circulatory system can trigger vasoconstriction, hypertension, aldosterone secretion, TGF-β-mediated inflammation, and fibrosis, and ultimately cardiac hypertrophy [144]. These symptoms closely resemble those exhibited by HFpEF in humans. Furthermore, in addition to causing obesity, a high-fat diet (HFD) is also known to induce a host of cardiac-related symptoms, including left ventricular hypertrophy, HFpEF, and diastolic dysfunction [145,146,147,148]. In addition, Withaar et al. [149] constructed a model with HFD and ANGII treatment, which exhibited higher levels of cardiac fibrosis as well as more severe diastolic dysfunction and cardiac hypertrophy compared to the single-treatment group with HFD or ANGII. Also, Schiattarella et al. [150] developed a model in which both HFD and the constitutive NO synthase inhibitor N^ω^-nitro-L-arginine methyl ester (L-NAME) were imposed. Although L-NAME caused an increase in diastolic and systolic blood pressure, the HFD-L-NAME group exhibited more significant cardiomyocyte hypertrophy and a reduction of myocardial capillary density.

### 3.4. Model of Aging Liver

The major liver disorders in relation to aging include alcoholic liver disease (ALD), non-alcoholic fatty liver disease (NAFLD), and liver fibrosis, with NAFLD becoming one of the most widespread CLDs and an indication of the need of a liver transplant in recent years [151]. NAFLD is marked by excessive fatty deposits within the liver, and some patients may develop non-alcoholic steatohepatitis (NASH), which is a more aggressive form of the disease with histological manifestations including steatosis, hepatocellular swelling, and lobular inflammation, eventually leading to liver fibrosis, cirrhosis, and even liver cancer [152].

Studies have shown that a HFD represents a major contributor to the onset and progression of obesity and its associated metabolic diseases [153]. Therefore, in addition to being used for HFpEF modeling, HFD can also be used for the modeling of mouse models of NAFLD. Mice fed with HFD for a prolonged period of time develop insulin resistance, hepatic steatosis, inflammation, and liver fibrosis [154]. In addition to HFD, diet-induced models of NAFLD include high fructose diet, high cholesterol and bile salt diet, methionine- and choline-deficient diet, as well as a choline-deficient, L-amino acid-defined (CDAA) diet [155,156,157,158]. Among them, Keisuke et al. [159] established a NASH model for the CDAA diet combined with intraperitoneal injection of lipopolysaccharide (LPS) in C57BL/6J mice. This model showed more severe NASH-associated pathologic phenotypes and significant NF-κB activation compared to the mice fed with the CDAA diet only. In addition, certain chemical drugs can be used to induce NAFLD in mouse models, including streptozotocin, carbon tetrachloride (CCl_4_), and tetracycline. CCl_4_, for example, can accelerate the process of steatosis and fibrosis by generating reactive oxygen radicals, resulting in the damage of hepatocyte structure and function, while metabolites of CCl_4_ can facilitate the release of pro-inflammatory cytokines, further aggravating liver injury [160]. However, the treatment of chemical drugs is often combined with dietary induction, as Kubota et al. [161] administered CCl_4_ subcutaneously eight times to C57BL/6N mice fed with HFD, and showed that mice in the HFD-CCl_4_ group exhibited more significant steatohepatitis compared to mice fed with only HFD. There are also genetic models of NAFLD, such as *ob*/*ob* (leptin deficient), *db*/*db* (leptin receptor deficient), and melanocortin receptor 4 knockout (*Mc4r**^−/−^*) mice. As with chemical induction, these genetically engineered mice usually require a combination of dietary induction to better model NAFLD [154]. For example, Mc4r is a G protein-coupled receptor expressed in the hypothalamic nucleus and is associated with modulation of food intake and metabolism [162]. *Mc4r**^−/−^* mice fed with HFD exhibit significant hepatic fibrosis with histological features of NASH similar to humans, i.e., inflammatory cell infiltration and hepatocyte ballooning [163].

### 3.5. Model of Chronic Kidney Disease

Chronic kidney disease (CKD) is clinically recognized as a premature aging disease that causes progressive systemic inflammation, vascular disease, muscle atrophy, and organismal weakness [164]. In recent years, the morbidity and mortality of CKD have been increasing year by year. Research in human and various animal models have shown that CKD exhibits features of cellular senescence, like significantly higher levels of p16 and p21 in CKD patients in comparison with age-matched healthy groups. Furthermore, high levels of p21 predict a poor prognosis in patients with chronic renal failure [165]. This implies a close relationship between CKD and cellular senescence.

Diabetic nephropathy (DN) serves as an important cause of CKD. Various mouse models associated with diabetic nephropathy are available, yet so far none of them can fully mimic human signs of the disease [166]. Many attempts have been made to better reproduce the signs of human DN. For example, *db*/*db* DBA/2J was used to establish a DN model, which exhibited glomerulosclerosis, loss of pedicles, and thickened glomerular basement membranes after growing to 12 weeks of age. Also, the *db*/*db* DBA/2J mice showed markedly higher albumin and albumin-to-creatinine ratio (ACR) in urine in comparison with *db*/*db* BLKS/J mice [167]. Focal segmental glomerulosclerosis (FSGS) is a widespread primary glomerular disease marked by podocyte impairment and loss, along with significant proteinuria [168]. Maimaitiyiming et al. [169] administered a single high dose of adriamycin to C57BL6/J mice, which induced signs such as proteinuria, glomerulosclerosis, and increased levels of inflammation. In addition, some data suggested that acute kidney injury (AKI) can increase the risk of CKD in patients [170]. Therefore, researchers have also developed mouse models for the transition from AKI to CKD to investigate the underlying mechanisms. For example, 129S1/SVlmj mice were modeled with AKI and CKD using a single low-dose injection of cisplatin (CP) in combination with a hyperphosphate diet, resulting in the development of CKD signs such as low creatinine clearance, kidney interstitial fibrosis, hyperphosphatemia, and vascular calcification [171]. In addition, Wei et al. [172] established a model of bilateral ischemia-reperfusion injury to CKD using C57BL/6 mice that exhibited incomplete recovery from AKI followed by a progressive decrease in glomerular filtration rate, elevated plasma creatinine, and CKD-related histopathological changes, including bilateral interstitial fibrosis and worsening of proteinuria.

### 3.6. Model of Osteoarthritis

Osteoarthritis (OA), the most prevalent type of arthritis, is a synovial joint disorder characterized with cartilage degeneration and osteophytes [173]. Osteoarthritis can have a variety of causes, including obesity, inflammation, trauma, or genetic factors, the most important of which is aging [174]. Aging of the body induces senescence in a variety of osteoarthritis-related cells, including osteoblasts, bone lining cells, cartilage cells, and bone marrow cells, etc. The development of osteoarthritis can be further promoted by various pro-inflammatory or pro-aging SASP factors secreted by certain senescent cells. These studies imply an inextricable relationship between osteoarthritis and aging of the body [175,176].

The spontaneous model is the simplest model for establishing OA in mice, i.e., no treatment is applied to the mice and they are allowed to develop OA spontaneously with age. The progression of OA in these mice is very similar to the progression of non-traumatic OA in humans, mimicking the natural wear and tear of the joint throughout its life course [177]. However, not all wild-type mice develop OA, and even when they do, they require a longer rearing period, so spontaneous OA models in mice are often combined with transgenic means. For example, the STR/ort mouse is one of the commonly used spontaneous OA models, which has a high incidence of OA because it often exhibits bone dislocation, elevated levels of oxidative stress, and elevated expression of various inflammatory factors including IL-1β, IL-12, and macrophage inflammatory protein-1β, and can develop OA spontaneously early in life [178,179]. Surgical induction is another commonly used method to induce OA in mice, which allows for the induction of OA at a specific joint. The most common surgical approach is anterior cruciate ligament transection (ACLT), which causes instability of the ACLT-treated joint and induces post-traumatic osteoarthritis (PTOA), mimicking the progression and pathogenesis of PTOA in humans. McCulloch et al. [180] used a combination of destabilized medial meniscotibial ligament (DMM) and cartilage scratch to model PTOA in C57BL/6 mice, and showed that mice that underwent both procedures exhibited more severe osteophytes and synovitis than mice in the single procedure treatment group. Furthermore, meniscectomy and oophorectomy are also commonly used to model OA in mice [179]. In addition, OA can be induced in mice by chemical treatment, of which collagenase is most commonly used, and the administration of collagenase to the kneecap of mice leads to a series of osteoarthritis-like lesions caused by patellar malalignment [181]. Another method of mouse OA modeling has been developed in recent years, the non-invasive mechanical load model, which has the advantage of no surgery requirement, thus avoiding the potential artefact of surgical intervention, infection, or variation due to surgery or healing. However, the age, sex (hormonal status), and strain of the mice may have a significant impact on the modeling results of such methods [182,183]. For example, using an indenter, researchers have applied steady pressure to the tibial plateau of mice, which caused damage to the synovium, meniscus, ligaments, and articular cartilage, thus triggering histopathological changes in PTOA such as articular cartilage loss, chondrocyte disintegration, meniscal hyperplasia, and mineralization [183].

### 3.7. Model of Aging Lung

Another serious problem associated with the increase in the aging population is the rise in morbidity and mortality from aging-related lung diseases, of which chronic obstructive pulmonary disease (COPD) and idiopathic pulmonary fibrosis (IPF) are the main types [184,185]. In humans, COPD includes chronic obstructive bronchitis with fibrosis and small airway obstruction, and emphysema with airspace enlargement, lung parenchymal destruction, loss of lung elasticity, and small airway closure [186]. In the case of IPF, increased deposition of ECM in the pulmonary interstitium is manifested, ultimately resulting in damage to the lung structure and function [185]. For COPD, smoking is the main cause, while the etiology of IPF is not yet clear. Taken together, we still know very little about the mechanisms in the development of both diseases. To address this issue, more established mouse models for the two aging-related lung diseases are available.

COPD mouse models have been established mainly by drug induction. Among them, cigarettes, as the primary contributor of COPD, are also most commonly used for the induction of mouse COPD models [187]. Chronic cigarette smoke (CS) exposure can be divided into two types, airway-only or systemic exposure, both of which can lead to COPD signs such as respiratory impairment, emphysema, small airway and vascular remodeling, or pulmonary hypertension in the lungs of mice [188]. However, this method requires a long modeling time, often lasting several months, leading to the development of some short-term mouse models of CS-induced COPD. For example, He et al. [189] used a 28-day CS exposure in combination with intraperitoneal injection of CS extract in C57BL/6 mice, which not only shortened the modeling time, but also showed comparable inflammatory levels and pathophysiological changes to long-term CS modeling. In addition, elastase is also widely used for COPD modeling. The most common model utilizes porcine pancreatic elastase, which mimics the release of neutrophil-derived elastase during COPD, leading to alveolar tissue rupture, emphysema, and further inflammatory responses. These models have low cost and short modeling time compared to CS exposure [190,191]. Furthermore, LPS is also commonly used in the induction of COPD models in mice [192,193]. Mebratu et al. [194] treated C57BL/6 mice with intranasal LPS and elastase, which resulted in COPD signs such as mucocyte hyperplasia and emphysema. Additionally, LPS and CSE co-treatment was also shown to significantly enhance the nuclear translocation of NF-κB, thereby promoting the inflammatory response in COPD model mice [195].

For mouse models of IPF, the most common method is the transtracheal administration of bleomycin (BLM), a chemotherapeutic agent derived from *Streptomyces verticillis*, which also has the side effect of causing acute lung damage and fibrosis in humans. BLM can chelate with metal ions such as iron ions and react with oxygen to produce superoxide, which can cause DNA breakage. On the other hand, BLM can also induce lipid peroxidation, leading to tissue damage. Such a process would ultimately result in severe inflammatory response and pulmonary fibrosis [196,197]. Moreover, BLM induces activation of the transcription factor NF-κB, which to some extent mediates the enhancement of pulmonary inflammation in BLM-treated mice [198,199]. A single transtracheal administration of BLM triggers acute lung injury and neutrophil-driven inflammatory response that lasts seven to ten days and changes to a fibrotic response by approximately day 14 after administration, with fibrosis levels peaking between 21–28 days [196,200]. However, there are some limitations to this modeling approach, most notably, the spontaneous regression of the fibrotic phenotype in mice modeled in this way after 28 days of modeling. To address this issue, many studies have used multiple-dose BLM administrations. For example, Redente et al. [201] developed a fortnightly three-dose BLM drip model, which showed a more significant fibrosis phenotype in comparison with mice treated with a single dose, and exhibited persistent and progressive characteristics. Similar to human IPF, this model produced a phenotype of alveolar epithelial cell remodeling. In addition to BLM, fluorescein isothiocyanate (FITC), asbestos, and silica can also be used for modeling pulmonary fibrosis, but the histopathological features of these pulmonary fibrosis models generated using these substances differ significantly from those of human IPF. For instance, FITC-induced pulmonary fibrosis does not produce fibroblastic lesions, while lung diseases induced by asbestos and silica are more similar to human asbestosis and silicosis, respectively [200]. One of the most commonly used methods for non-chemical-induced IPF is radiation induction [202]. A single exposure to radiation can induce the fibrotic process by triggering DNA breaks that lead to alveolar epithelial cell death and tissue damage, ultimately causing inflammation and fibrotic responses in the lung. Nevertheless, it should be noted that this modeling method takes longer and also lacks some of the histopathological features of human IPF [203]. In addition, some transgenic mice can be used for modeling pulmonary fibrosis. For example, the surfactant protein C (SP-C) gene (*Sftpc*) is expressed only in alveolar epithelial type II cells (AEC2s), and mutations in this gene are thought to be associated with familial interstitial lung disease [204,205]. Nureki et al. [206] selectively expressed human mutant SP-C within AEC2s in mice, which led to diffuse lung injury, and subsequently elevated levels of lung inflammatory response and spontaneously remodeled lung fibrosis. In addition, mice with defects in some telomere-related genes have been used for modeling pulmonary fibrosis, such as *Tert* [207] and *Trf1* [208] deficient mice. However, it has also been previously shown that although telomerase deficiency leads to telomere shortening in mice, there is no difference in the phenotype of pulmonary fibrosis produced by wild-type and telomere deficient mice treated with BLM at the same time [209]. Therefore, further studies are needed to determine whether telomere-associated protein defects can cause or promote the development of pulmonary fibrosis. Mucin 5B is encoded by the *Muc5b* gene and has an essential function in mucociliary clearance and host defense. Hancock et al. [210] demonstrated that the presence of excessive *Muc5b* within mouse AEC2s enhanced the level of BLM-induced pulmonary fibrosis. Further, Nedd4-2, a ubiquitin ligase, can be engaged in a variety of cellular processes related to epithelial homeostasis, and its dysregulation might correlate with the development of chronic lung disease and fibrosis. Studies have shown that conditional knockout of Nedd4-2 in epithelial cells can lead to pulmonary fibrosis [211,212]. Finally, overexpression of some cytokines in mice can also be applied to model pulmonary fibrosis. For example, adenovirus-based delivery of TGF-β1 in mice leads to alveolar collapse associated with surfactant dysfunction, which further results in the development of fibrosis [213].

## 4. Discussion

In this review, we summarize the accelerated aging mouse models, both in systematic whole-body and in specific organs. Although systemic-induced accelerated aging mouse models mimic natural aging, it is not fully equal to natural aging. Models of aging organs can mimic specific diseases to some extent while not representing the overall aging process in the body.

When assessing the effects of inducing an accelerated aging model with the markers of aging, not all markers show significant or consistent differences and some even exhibit contradictory results. Therefore, diverse mouse models have been employed to recapitulate different features of diseases, which implies the complexity of the pathogenesis of various illnesses. Accordingly, multiple and diverse mouse models are necessary to elucidate the pathogenesis of the various disorders that develop during human aging.

For instance, *Lmna* mutant mice (i.e., *Lmna^L530P/L530P^* and *Lmna^G609G/G609G^* mice) can serve as a model for accelerated aging, but the cardiovascular phenotype exhibited by such mice is far from the cardiovascular alterations of humans that occur during natural aging. While cardiomyocytes expand and arterial walls thicken with age in humans, *Lmna* mutant mice exhibit cardiomyocyte atrophy and depletion in vascular smooth muscle of the aortic arch [214]. Moreover, the D-galactose-induced premature aging model can be highly variable among rodents of different species or ages, e.g., intraperitoneal administration of 300 mg/kg D-galactose for two months to four-week-old Wistar rats did not influence their anxiety levels, spatial cognition, memory, or neurogenesis [215].

Due to the space limit, we may miss some aging mouse models. The development process has been proved to affect the aging process. For example, intrauterine growth retardation can cause the offspring to have an increased incidence of type 2 diabetes in adulthood, thus affecting their lifespan. These mouse models can also be a premature senescence model for energy metabolism studies. Low protein restriction, total calorie restriction, and maternal glucocorticoid exposure to the pregnant rodents are commonly used to induce intrauterine growth retardation. Also, induction of uteroplacental insufficiency by a uterine artery ligation surgery can limit the nutrient supply to the fetus. These treatments lead to epigenetic reprogramming to offspring in utero, causing their predisposition to diabetes and other metabolic disorders in adults [216].

It should be noted that accelerated aging is not the same as natural aging. The stimuli induced by the above induction methods can trigger adaptive responses in mice, and when these adaptive responses are exhausted, the failure of various tissues and organs in mice is induced. The various aging phenotypes produced by this process are quite different from those of natural aging and the accelerated aging models cannot fully reflect the pathogenesis of various aging-related diseases in humans. Taking the various AD mouse models mentioned above as examples, although some transgenic models can display the histopathological features of AD, the pathogenesis of AD not only involves mutated genes such as *APP* and *PSEN*, but also environmental factors (e.g., heavy metals) or chronic stress [125,217]. It is also worth noting that the organ background of the mice themselves has an important influence on the phenotypic outcome of the various models, which is particularly reflected in the modeling results in mice of different ages. For example, in the BLM-induced pulmonary fibrosis model, older mice were more susceptible to stimuli than younger mice and showed a more pronounced fibrotic phenotype and higher TGF-β expression in their lungs, exhibiting less spontaneous regression of fibrosis [218,219]. Therefore, when conducting aging-related disease studies, we must select mice of an appropriate age or genetic background for disease modeling according to the study objectives. Because of ethics-related issues, long lifespan, environmental effects, and various constraints, it will remain difficult to study humans as subjects for aging and its relevant disorders, at least for the foreseeable future. In this review, we summarize some of the commonly used and emerging aging-related mouse models in recent years. We found that although mice are the most commonly used experimental animals today, and the utilization of mouse models has contributed to our understanding of pathogenesis of aging and related diseases to a certain extent, there are few mouse models that can well replicate the aging process or the development of aging-related diseases in humans, which has seriously hindered the progress of research in this field. The methods of induced aging may require further research and improvement, for instance, exploring the possibility of combined effects. As the research progresses, we will hopefully get more positive results.

## Figures and Tables

**Figure 1 cells-11-01418-f001:**
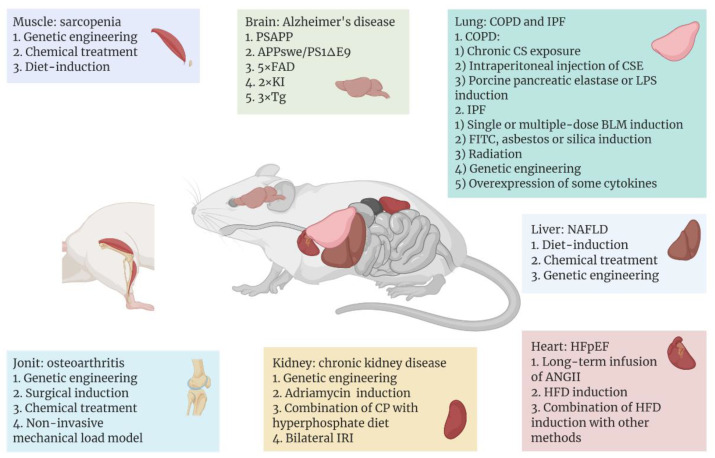
Aging-related diseases include Alzheimer’s disease (AD), sarcopenia, heart failure with preserved ejection fraction (HFpEF), non-alcoholic fatty liver disease (NAFLD), chronic kidney disease (CKD), osteoarthritis, chronic obstructive pulmonary disease (COPD), and idiopathic pulmonary fibrosis (IPF), which occur in different tissues or organs. With the help of mice as model organisms, researchers have used different methods to establish disease models in different tissues or organs of mice. In recent years, some common or emerging methods of modeling aging-related diseases are shown above, and these methods are classified according to the organ to which the disease mainly affects. Figure 1 was created with BioRender software (https://biorender.com/ (accessed on 20 April 2022)).

**Table 1 cells-11-01418-t001:** Summary of systemic-induced accelerated aging mouse models.

Type	Subdivision	Phenotypes
D-galactose-induced senescence model	Brain	Cognitive impairment Mitochondrial dysfunction Neuronal degeneration Apoptosis Depressive and anxious
	Heart	Cardiac fibrosis Collagen accumulation Fibroblasts disordered arrangement
	Kidney	Kidney index ↓ Uric acid & Cys-C ↑ Glomerular and tubular damage ↑
	Liver	Liver fibrosis Glycogen levels ↓ Lipid deposition ↑
	Reproductive system	Estrogen and progesterone ↓ Ovarian follicle regression Uterine wall endometrial gland atrophy Disrupt estrous cycles
	Intestinal flora	Disturbance
	Lung	Oxidative stress ↑ Fibrotic status Chronic inflammation
SAMP mice	SAMP 1	Aging amyloidosis Immune dysfunction Renal atrophy Hearing loss Senile pulmonary hyperinflation
	SAMP 6	Senile osteoporosis Myeloid progenitor cell senescence
	SAMP 8	Astrogliosis Microgliosis Neurodegeneration Amyloid accumulation MAPT hyperphosphorylation
	SAMP 10	Learning and memory impairment Cerebral cortex and limbic system atrophy
Rps9 D95N mouse		Altered fur Cataracts Hunched posture Body composition function & body weight ↓ Fat mass & muscle strength ↓ Shortened lifespan Mouse urinary syndrome Extramedullary hematopoiesis
	*Lama^−/−^*	Short lifespan Growth retardation Muscular dystrophy Altered lipid metabolism
	*Wrn^∆hel/∆hel^*	Short lifespan Abnormal hyaluronic acid excretion Metabolic abnormalities Increased genomic instability and cancer incidence
	*Csa^−/−^*, *Csb^−/−^*	Short lifespan Reduced fat mass Photoreceptor cell loss Neural pathology
Progeria syndrome mouse	*Xpd^TTD/TTD^*	Short lifespan Trichothiodystrophy
	*Bub1b^H/H^*, *Xpg^−/−^*	Brain atrophy Neuronal loss Neurofibrillary deposition of Aβ or senile plaques
	*Bub1b^H/H^*, *Bub1b^+/GTTA^*	Mean muscle fiber diameter ↓ Muscle fiber size variation ↑ Intermuscular fibrosis Regenerative capacity of skeletal muscles ↓
Mitochondrial DNA polymerase mutant mouse		Lifespan ↓ Weight loss Subcutaneous fat ↓ Hair loss Kyphosis Osteoporosis Anemia Fertility ↓ Spermatogonia depletion Heart enlargement
Total body irradiation (TBI) model		Progressive premature frailty Cognitive decline Whole blood antioxidant capacity ↓ RBC glutathione ↓ Thymic involution Articular cartilage and bone degeneration Ovarian environment damage
Ozone-induced senescence model		Cognitive decline Memory impairment AD symptoms Lung tumor growth ↑
Chronic jet-lag mouse		Accelerated initial tumor growth Shortened mouse survival Induce spontaneous hepatocellular carcinoma Obesity Depression Addiction Abnormal cardiac structure Impaired cardiac function

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
