# Peer review of "Induction of Accelerated Aging in a Mouse Model"

_cells, 2022, doi:10.3390/cells11091418_

Round 1

Reviewer 1 Report

This is an excellent review and I enjoyed reading it.

I have few comments:

1) Please, check the english for some spelling and grammar mistakes.

2) In general, "models of accelerated aging" is best wording to characterize the review. This is particularly important in the second part of the review when presenting specific organs: heart, liver, muscle, kidney, etc, are all models of diseases and that they end-up expressing markers of aging is after all normal. Yet, as you started in the discussion, these models are specific models, addressing a specific organ, likely not representative of aging per se.

3) One category is missing and would benefit the review to be introduced: the developmental origin of diseases and aging. It is a large subject but it is becoming increasingly prevalent. Intrauterin environment has profond effects on organ development and, because of the global impact of the fetal development, these models may be the best models of premature aging.

4) in the discussion, I would put even more emphasis on the fact that accelerating aging is not equal to natural aging. Stresses, what ever they are, induce a defense response: this defense response itself will induce an adaptive response of the bodily functions. When adaptive responses are exhausted, there is failure, best representated by heart, kidney or brain failure. In addition, the AD mouse models are based on genetic defects: in humans, these genetic defects only represent a tiny percentage of what is considered now sporadic AD in the elderly: I do not believe that these models (APP, preseniline,..) are representative of sporadic AD in humans. I would make sure in the discussion as well as in the introduction, that these models, although useful, have strong limitations that have to be considered in all interpretations. For example, inducing NASH in a young animal is unlikely to be equal to inducing NASH in an 18-month old mouse. Responses are going to be different because the organ background will be different. This should be acknowledge. I am from the cardiovascular field and for example, starting an antioxidant treatment at 3-mo and for 9 months is deleterious, while starting the same at 9-mo is protective on the endothelial function; now, if the mouse is genetically dyslipidemic, it is the opposite! this is a point that is essential and I believe you could discuss this for the benefit of all readers.

Author Response

1. Please, check the english for some spelling and grammar mistakes.

Answer: Thanks for your suggestion. We have revised a few spelling and grammar mistakes.

2. In general, "models of accelerated aging" is best wording to characterize the review. This is particularly important in the second part of the review when presenting specific organs: heart, liver, muscle, kidney, etc, are all models of diseases and that they end-up expressing markers of aging is after all normal. Yet, as you started in the discussion, these models are specific models, addressing a specific organ, likely not representative of aging per se.

Answer: We agree with you. We have revised the discussion to emphasize this point.

3. One category is missing and would benefit the review to be introduced: the developmental origin of diseases and aging. It is a large subject but it is becoming increasingly prevalent. Intrauterin environment has profond effects on organ development and, because of the global impact of the fetal development, these models may be the best models of premature aging.

Answer: Thanks for your suggestion. We have added some premature senescence models for energy metabolism studies in the 4th paragraph of the discussion.

4. in the discussion, I would put even more emphasis on the fact that accelerating aging is not equal to natural aging. Stresses, what ever they are, induce a defense response: this defense response itself will induce an adaptive response of the bodily functions. When adaptive responses are exhausted, there is failure, best representated by heart, kidney or brain failure. In addition, the AD mouse models are based on genetic defects: in humans, these genetic defects only represent a tiny percentage of what is considered now sporadic AD in the elderly: I do not believe that these models (APP, preseniline,..) are representative of sporadic AD in humans. I would make sure in the discussion as well as in the introduction, that these models, although useful, have strong limitations that have to be considered in all interpretations. For example, inducing NASH in a young animal is unlikely to be equal to inducing NASH in an 18-month old mouse. Responses are going to be different because the organ background will be different. This should be acknowledge. I am from the cardiovascular field and for example, starting an antioxidant treatment at 3-mo and for 9 months is deleterious, while starting the same at 9-mo is protective on the endothelial function; now, if the mouse is genetically dyslipidemic, it is the opposite! this is a point that is essential and I believe you could discuss this for the benefit of all readers.

Answer: Thanks for your wonderful suggestion. We have these points and examples in the 5th paragraph of the discussion.

Reviewer 2 Report

Even though author mentioned that the natural aging model is time consuming, labor-intensive, expensive, and the effect of aging is not stable, I think natural aging model best mimics the real scenario in human aging cases.

It would be great if you could add natural aging model study examples.

Author Response

Comments and Suggestions for Authors Even though author mentioned that the natural aging model is time consuming, labor-intensive,expensive, and the effect of aging is not stable, I think natural aging model best mimics the real scenario in human aging cases.

It would be great if you could add natural aging model study examples.

Answer: Thanks for your wonderful suggestion and we agree with you. Considering that the aging process is dynamic and complex, it is difficult for a single drug-induced aging method to fully simulate the natural aging process. Natural aging mouse model best mimics the phenotypes in human aging process. We add some examples in the introduction. We have added the natural aging models such as naked mole-rats in the 3rd paragraph of the introduction.

Reviewer 3 Report

The study is really good and it is a nice review about the accelerated aging in mouse model. I have only two comments:

Comment 1. In the line 37, the authors wrote that the effect of aging is not stable, they should explain better the stable concept.

Comment 2. In the manuscript, the authors should specify the factor of NF-kB is implicated in the several discussed models

Author Response

Comment 1. In the line 37, the authors wrote that the effect of aging is not stable, they should explain better the stable concept.

Answer: Thanks for your suggestion. We feel the term “stable” is not appropriate thus we have deleted this.

Comment 2. In the manuscript, the authors should specify the factor of NF-kB is implicated in the several discussed models

Answer: Thanks for your suggestion. We have revised and specified the factor of NF-kB in these models. NF-κB is an important regulator of the inflammatory response and its activation promotes the gene expression of a variety of cytokines. The onset and progression of many aging-related diseases (e.g. Alzheimer's disease, osteoarthritis, non-alcoholic steatohepatitis, idiopathic pulmonary fibrosis, etc.) are often accompanied by the development of inflammatory responses. According to previous studies on human tissue samples, NF-κB activation is an important contributor to such inflammatory responses and has important implications for the pathogenesis of these aging-related diseases. The same results have also been validated in mouse models. For example, it has been shown that treatment with BLM activates NF-κB signaling in the lungs of mice and that further improvement of pulmonary fibrosis symptoms is achieved when mice are treated with inhibitors of NF-κB.